**Subject Category:**
Biology (whole organism)

biomechanics

frog, three-dimensional kinematics, locomotion, pelvis, Hyperoliidae

**Author for correspondence:**
Amber J. Collings
e-mail: a.collings@tees.ac.uk

# The impact of pelvic lateral rotation on hindlimb kinematics and stride length in the red-legged running frog, *Kassina maculata*

Amber J. Collings[1,2], Laura B. Porro[3], Cameron Hill[2] and Christopher T. Richards[2]

[1]School of Science Engineering and Design, Teesside University, Middlesbrough TS1 3BX, UK
[2]Structure and Motion Laboratory, Royal Veterinary College, Hawkshead Lane AL9 7TA, UK
[3]Cell and Developmental Biology, University College London, London WC1E 6BT, UK

AJC, 0000-0002-6488-5085; CTR, 0000-0002-1908-3577

Some frog species, such as *Kassina maculata* (red-legged running frog), use an asynchronous walking/running gait as their primary locomotor mode. Prior comparative anatomy work has suggested that lateral rotation of the pelvis improves walking performance by increasing hindlimb stride length; however, this hypothesis has never been tested. Using non-invasive methods, experimental high-speed video data collected from eight animals were used to create two three-dimensional kinematic models. These models, each fixed to alternative local anatomical reference frames, were used to investigate the hypothesis that lateral rotation of the mobile ilio-sacral joint in the anuran pelvis plays a propulsive role in walking locomotion by increasing hindlimb stride length. All frogs used a walking gait (duty factor greater than 0.5) despite travelling over a range of speeds (0.04–0.23 m s$^{-1}$). The hindlimb joint motions throughout a single stride were temporally synchronized with lateral rotation of the pelvis. The pelvis itself, on average, underwent an angular excursion of 12.71° ($\pm$4.39°) with respect to the body midline during lateral rotation. However, comparison between our two kinematic models demonstrated that lateral rotation of the pelvis only increases the cranio-caudal excursion of the hindlimb modestly. Thus, we propose that pelvic lateral rotation is not a stride length augmenting mechanism in *K. maculata*.

# List of symbols and abbreviations:

| | |
|---|---|
| COM | Centre of mass |
| FBA | Fixed body axis |
| FPA | Fixed pelvic axis |
| IS | Ilio-sacral |
| LMM | Linear mixed-effects model |
| **M** | Matrix |
| $\mathbf{M_{BA}}$ | Matrix of body axis corrected coordinates |
| $\mathbf{M_{PV}}$ | Matrix of pelvic vector rotation corrected coordinates |
| $\theta_{BA}$ | Body axis angle; angle between body axis line and $x$-axis |
| $\theta_{PV}$ | Pelvic vector angle; angle between pelvic vector axis line and $x$-axis |
| s.d. | Standard deviation |
| TMT | Tarsometatarsal |
| $\mathbf{v_{Bod}}$ | Body axis vector; straight line between head and vent marker |
| $\mathbf{v_{Pel}}$ | Pelvic vector; straight line between sacrum and vent marker |

# 1. Introduction

Anurans are recognized for their jumping and swimming abilities; however, walking is thought to be a key behaviour present since the origin of the group [1]. Among modern frogs, many species have secondarily acquired walking and running ability as their principal locomotor mode [1,2]. Accordingly, biomechanists have investigated walking/running frogs to expand our fundamental understanding of quadrupedal vertebrate locomotion [3,4] as well as to link behavioural traits to morphology [5], muscular function [6] and energetics [7,8]. Yet, despite the growing interest, studies on walking/running are sparse compared to the vast literature on jumping and swimming [9–23]. Currently, while there is strong evidence explaining the fundamental mechanism of jumping [24] and swimming [22,25,26], we lack complete mechanistic insight into anuran quadrupedal locomotion.

Our current knowledge of frog walking derives from recent observational studies [4,5]. Among several species tested, frogs walk by maintaining each limb in ground contact for at least 50% of the stride duration (i.e. duty factor $\geq$50% [4]) and show remarkably similar footfall and limb kinematics across speed and species [5]. Additionally, frogs employ a quadrupedal limb contact cycle where the forelimb (on the leading side) makes contact first, followed by the leading hindlimb then the trailing hindlimb and, finally, the trailing forelimb [5]. This contact sequence, known as the *lateral sequence* [27] is common among many quadrupedal taxa [28], including walking salamanders (but not trotting [29]).

Because salamanders are the closest related extant taxa to anurans [1], comparative morphologists have used salamander walking as a template to infer how frog limbs and their body axis contribute to propulsion [30]. The body axis in walking salamanders bends to allow the pelvic girdle to sweep in an arc in the horizontal plane. This axial bending increases stride length, bringing the hand and the foot of the same side closer together [29,31].

As a starting point, we assume *a priori* that anurans also employ lateral axial bending to maximize stride length. In contrast to salamanders and other sprawled-limb quadrupeds, however, frogs possess a short, stiff spine [30]. As a solution, frogs possess a hinge-like mechanism called the ilio-sacral (IS) joint [32] which originated *de novo* early in frog evolution [33]. Rotation about the IS joint during walking is driven by pelvic and spine muscles that are activated reciprocally (alternating left–right firing) to cause medio-lateral pelvic rotation [6]. However, it is currently unknown to what extent the IS joint contributes to increasing stride length in anurans which habitually use walking as their main form of locomotion.

Using a combination of experimental kinematics and modelling, we aim to quantify pelvic lateral rotation and test the hypothesis that the IS joint plays a propulsive role in frog locomotion by increasing stride length. Following prior studies on walking kinematics [4,5], we use *Kassina maculata* as our model species, a terrestrial Hyperoliid frog renowned for its walking/running gait [4]. Our novel kinematics modelling approach uses two models, each fixed to alternative local anatomical reference frames. The first model provides a quantitative assessment of the extent of pelvic rotation, whereas the second model uses an alternative reference frame to mathematically fix pelvic rotation

and measure hindlimb excursion without the contribution of pelvic lateral rotation. A comparison between the two models then allows an estimation of pelvic contribution to hindlimb stride length. To our knowledge, the current study is the first to quantify three-dimensional walking kinematics analysing the integrated motion of the hindlimb and pelvis, testing long-standing assumptions about frog walking mechanics. Furthermore, the use of these models, based on experimental kinematic data, provides a novel and non-invasive method of studying the influence of pelvic lateral rotation on the hindlimb. Shedding light on the musculoskeletal function of the pelvic apparatus during walking, along with recent findings [4,5], may bring us closer to understanding the ways in which modern frogs overcome their jumping 'specializations' and provide insight to the driving forces behind the evolution of the unique anuran pelvic apparatus.

# 2. Material and methods

## 2.1. Animals

Data were collected from eight adults (17.87 ± 3.35 g body mass, 53.91 ± 16.50 mm snout–vent length; AmeyZoo, Hemel Hempstead, UK). The frogs were housed in shared pens kept at 23–25°C under a 12 : 12 hour light : dark cycle situated at the Royal Veterinary College, Biological Sciences Unit. Animals were provided shelter and access to a fresh water bath at all times and were fed live crickets three times weekly, except on the day of experimentation. All procedures were approved under the Home Office Licence 70/8242.

## 2.2. High-speed video data collection

Twelve circular skin markers (approx. 4 mm diameter) were created from thin (approx. 0.25 mm thick) plastic, painted white with a black dot in the centre (approx. 1 mm diameter). The markers were placed overlying the approximate position of skeletal landmarks of the right hindlimb and body (figure 1$a$,$b$) using cyanoacrylate adhesive. The frogs were briefly anaesthetized for the application of these markers, using a solution of 0.01% tricaine methanesulfonate (MS222), 0.02% sodium bicarbonate (NaHCO$_3$) and de-chlorinated water, and were left to recover for at least 30 min before the collection of video data.

Two high-speed video cameras (Photron SA3, Photron Ltd, USA) were used to record footage of frogs ($n = 23$ trials pooled across $N = 8$ frogs) walking freely along a moistened acrylic surface (250 Hz, 1/1500 shutter speed). The animals were allowed to travel at self-selected speeds to simulate natural walking locomotion and ensure ecologically relevant data. One camera provided a dorsal view while the other provided a lateral view. Cameras were calibrated using a 49-point custom-built calibration block. A mirror, positioned directly opposite the lateral camera, was used to obtain a third view. The mirror was mounted at an approximately 60° angle from the floor of the trackway such that all markers on the body and leg could be observed simultaneously in each frame. Immediately after filming, direct measurements of the body and hindlimb segment lengths were obtained using digital callipers, before the markers were gently removed (table 1).

Throughout data collection, an effort was made to capture at least one full stride (right hindlimb toe-on to toe-off) in the field of view. Acceptable trials were those in which the frog moved forward continuously without turning and completed at least one full stride, mid-locomotion (initial 'push-off' strides and final strides as they entered into a sitting position were rejected).

## 2.3. Defining events within the limb cycle

The transitions between swing and stance were neither clear nor instantaneous, lacking defined toe-on and toe-off events. Throughout the stance–swing transition, the frogs gradually peeled their feet off of the ground, while the transition from swing to stance was variable between forelimb and hindlimb. In the forelimb, the frogs placed each digit down sequentially from the medial to the lateral digit. This pattern was less clear in the hindlimb, however, where the foot gradually came into contact with the ground over several frames. Therefore, to best define the swing and stance phases, the following transition points were determined. The stance–swing transition point, for both the forelimb and hindlimb, was defined as the frame in which the digits had peeled away from the ground enough for the angle between the plantar surfaces of the digits and the ground

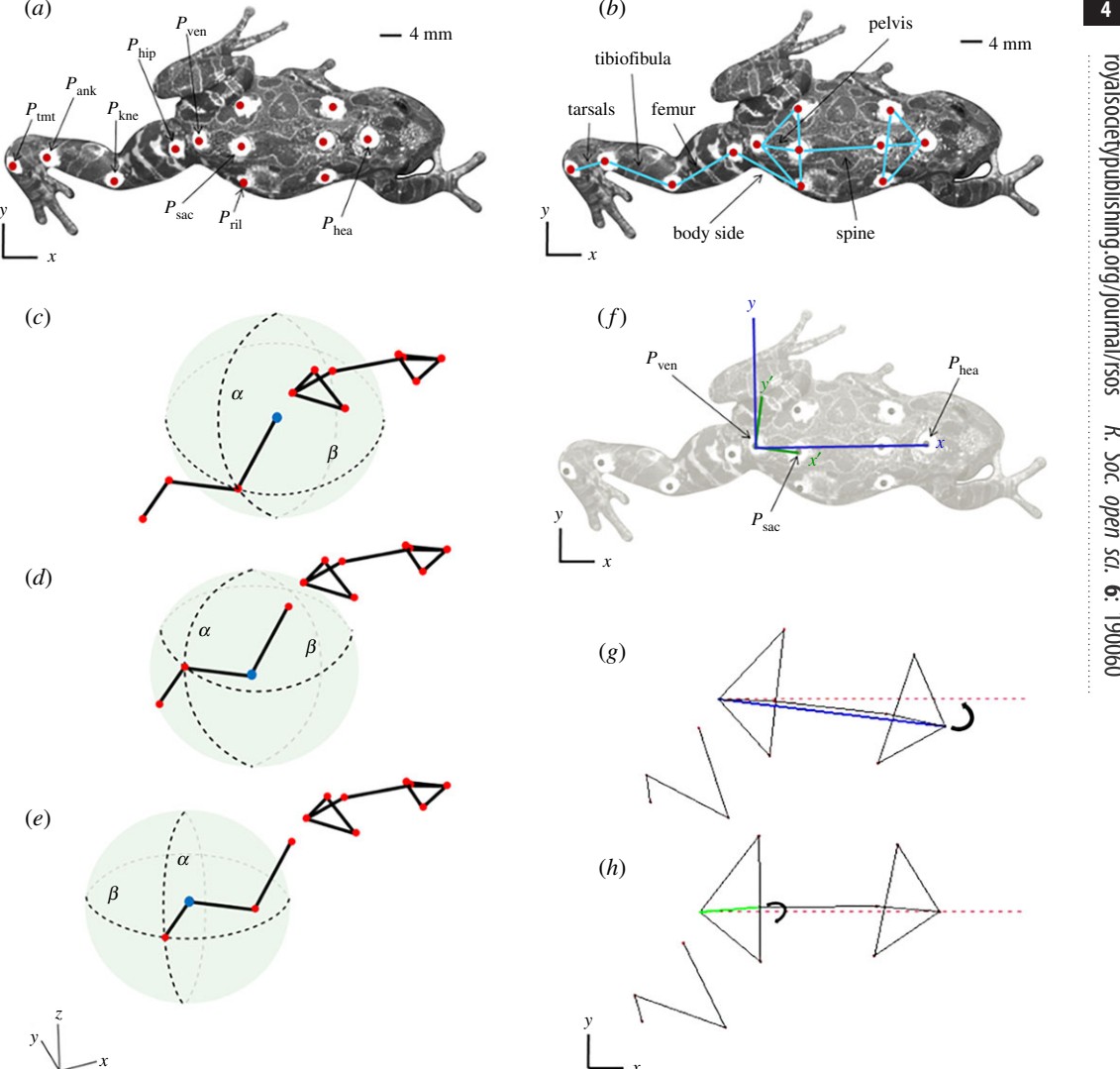

**Figure 1.** Data collection and model construction. Demonstration of the name and position of all external hindlimb, midline and body skin markers, from dorsal view (*a*), demonstration of skeletal and limb segments based on skin markers, from dorsal view (*b*), and a schematic detailing calculation of polar coordinates (*c–e*). Limb segment in question treated as the radius of a sphere, horizontal protraction/retraction of the limb segment results in motion along $\beta$, whereas vertical adduction/abduction results in motion along $\alpha$. (*c*) femur, (*d*) tibiofibula, (*e*) tarsals segment motion. Red markers represent limb joints, whereas the blue marker represents the limb joint being treated as the origin of the sphere. Demonstration of the local anatomical reference frames for the creation of the two alternative models (*f*). Body axis reference frame shown in blue, pelvic axis reference frame shown in green. A schematic for the visualization of rotation matrix applied to fix the body axis frame for the FBA model (*g*), and a schematic for the visualization of the rotation matrix applied to fix the pelvic vector to create the FPA model (*h*). Body axis vector in blue, pelvic vector highlighted in green, and dashed red line represents global X-axis. Animals viewed in global reference frames as shown.

to exceed approximately 90°. The swing–stance transition in the forelimb was defined as the frame in which the second digit made contact with the ground. This transition point in the hindlimb was defined as the first frame where over approximately 50% of the foot was in contact with the ground.

## 2.4. Spatio-temporal analysis of hindlimb joint three-dimensional kinematics

Video data were calibrated and digitized using Matlab (MathWorks, Natik, USA) Digitizing Tools DLTcal5 and DLTdv5 [34]. The digitization process produced three-dimensional coordinates defining the position of each marker, in each frame, in the capture volume. These coordinates were then loaded into MATHEMATICA 9.0.1.0 (Wolfram Research, Champaign, USA) where all further data

**Table 1.** Individual and pooled animal anatomical measurements. Mean values presented as mean ± s.d.

| animal number | animal morphology | | | hindlimb segment lengths (mm) | | | |
|---|---|---|---|---|---|---|---|
| | body mass (g) | SVL (mm) | body width (mm) | hip–knee | knee–ankle | ankle–TMT | TMT–digit |
| 1 | 23.47 | 62.55 | 35.21 | 15.67 | 20.05 | 9.04 | 24.09 |
| 2 | 17.53 | 55.55 | 33.07 | 13.92 | 18.14 | 11.61 | 20.08 |
| 3 | 16.05 | 59.83 | 30.78 | 15.40 | 21.56 | 11.77 | 22.10 |
| 4 | 21.08 | 57.60 | 34.22 | 18.74 | 21.18 | 11.42 | 26.33 |
| 5 | 14.55 | 47.38 | 32.22 | 13.95 | 19.28 | 11.34 | 19.49 |
| 6 | 13.68 | 45.06 | 28.39 | 14.26 | 17.54 | 9.35 | 19.46 |
| 7 | 19.82 | 56.03 | 32.07 | 18.41 | 14.56 | 10.56 | 24.20 |
| 8 | 16.76 | 47.25 | 16.08 | 21.05 | 20.10 | 12.54 | 20.30 |
| mean | 17.87 ± 3.35 | 53.91 ± 6.50 | 30.26 ± 6.09 | 16.43 ± 2.66 | 19.05 ± 2.28 | 10.95 ± 1.22 | 22.01 ± 2.60 |

processing and analysis was conducted. The X, Y and Z coordinates were filtered using a second-order reverse Butterworth filter with a cut-off frequency of 30 Hz. A dynamic stick figure model for each trial was produced using the marker coordinates for each trial. Straight lines connecting the digitized marker points were used to represent the limb segments.

Walking speed was calculated from the position of the head marker, in metres per second ($m\,s^{-1}$), between the start of stance and the end of the limb cycle. Stride length was calculated as the Euclidean distance between the position of the foot at the start of stance phase and the position of the same foot at the end of the limb cycle, using the longest digit of the right hindlimb for reference. Stride frequency was calculated as one over stride duration (seconds). Hindlimb motion was then characterized using two methods: the three-dimensional vector angle method and the polar coordinate angle method.

## 2.5. Three-dimensional vector angle method

The three-dimensional vector angle method was used to calculate the scalar angle between the hindlimb segments, represented as vectors, in each frame of the walking stride. To obtain the three-dimensional joint angle data from the kinematic model of the frog we: (i) mathematically tethered the model and (ii) calculated three-dimensional vector angles for the hip, knee and ankle joint. The model was mathematically tethered at the hip joint, achieved by calculating the difference between the hip joint marker coordinates and all other marker coordinates in each frame such that the hip became the origin (hip X, Y, Z = [0, 0, 0]) for all frames. Three-dimensional vector angles for the hip, knee and ankle joints were calculated in MATHEMATICA.

Firstly, the skin marker points ($P_{kne}$, $P_{hip}$, $P_{ril}$, $P_{ank}$, $P_{tmt}$) were used to define the joint segments (body side, femur, tibiofibula and tarsals; see figure 1*a*,*b* for anatomical positioning of the marker points and joint segments). The joint segments were then represented as vectors, achieved by calculating the difference between the proximal and distal marker points describing the segment in question. As such, the body side vector ($\mathbf{v}_{side}$) was calculated as the difference between the right ilia and the hip marker points, the femur vector ($\mathbf{v}_{fem}$) was calculated as the difference between the hip and the knee marker points, the tibiofibula vector ($\mathbf{v}_{tib}$) was calculated as the difference between knee and the ankle marker points, and the tarsals vector ($\mathbf{v}_{tar}$) was calculated as the difference between the ankle and the tarsometatarsal joint (TMT) marker points (equations (2.1)–(2.6)). Pairs of adjacent vectors spanning a joint were then used to calculate joint three-dimensional angles as per equations (2.1)–(2.6).

For the hip three-dimensional angle:

$$\mathbf{v}_{fem} = P_{kne} - P_{hip} \tag{2.1}$$

$$\mathbf{v}_{side} = P_{ril} - P_{hip} \tag{2.2}$$

For the knee three-dimensional angle:

$$\mathbf{v}_{fem} = P_{hip} - P_{kne} \tag{2.3}$$

$$\mathbf{v}_{tib} = P_{ank} - P_{kne} \tag{2.4}$$

For the ankle three-dimensional angle:

$$\mathbf{v}_{tib} = P_{kne} - P_{ank} \tag{2.5}$$

$$\mathbf{v}_{tar} = P_{tmt} - P_{ank} \tag{2.6}$$

Secondly, the three-dimensional vector angle between the limb segment vectors was computed using the vector angle function in MATHEMATICA. Equation (2.7) demonstrates the calculation of the hip three-dimensional angle as an example, all other joint angles were calculated similarly.

$$\theta\mathrm{Hip} = \mathrm{acos}\left(\frac{\mathbf{vfem} \cdot \mathbf{vside}}{\|\mathbf{vfem}\|\|\mathbf{vside}\|}\right) \tag{2.7}$$

where $\theta$Hip represents the hip three-dimensional vector angle.

## 2.6. Polar coordinate angle method

Whereas the three-dimensional vector angle method described in the previous section calculates the scalar angle between vectors without regard for orientation, the polar coordinate angle method decomposes joint motion into the flexion/extension and abduction/adduction components (figure 1c−e). To further characterize joint motion, polar coordinate analysis was used to assess the range of motion of each joint in the cranio-caudal direction and the dorsoventral direction. While Cartesian coordinate systems describe the position of a point in three-dimensional space using an X-, Y- and Z-value, polar coordinate systems describe the position of a point in a spherical three-dimensional space using two angles corresponding to longitude and latitude. To obtain polar coordinate angles for each of the limb segments (femur, tibiofibula and tarsals), as previously with the three-dimensional vector angle analysis, the segments were transformed into vectors by calculating the difference between the proximal and distal joint. This effectively rendered the proximal joint of the limb segment as the centre of the spherical three-dimensional space. The orientation of the distal joint was then transformed from Cartesian coordinates to polar coordinates in MATHEMATICA, such that the direction of the limb segment and position of the distal joint were described by a longitude and latitude angle (figure 1c−e). In this orientation, the longitude angle described abduction/adduction and the latitude angle described flexion/extension (protraction/retraction) of the limb segments. Equations (2.8)–(2.10) define the relationship between polar and Cartesian coordinates.

$$r = \sqrt{x^2 + y^2 + z^2} \tag{2.8}$$

$$\sigma = \mathrm{atan}\left(\frac{y}{x}\right) \tag{2.9}$$

$$\phi = \mathrm{acos}\left(\frac{z}{r}\right) \tag{2.10}$$

where $r$ is the radial coordinate describing the distance from the origin, $\sigma$ represents the flexion/extension and $\phi$ represents abduction/adduction.

In contrast to previous studies using joint-based coordinate systems [20], the above polar representation gives segment orientations in the frog's global reference frame. Unlike local coordinates, global orientation allows us to easily resolve how the motion of a given segment translates into frog body motion [35]. For example, large changes in $\sigma$ relative to $\phi$ indicate that a given segment is moving in the horizontal plane and probably contributing to forward thrust of the animal. Reciprocally, if changes in $\phi$ dominate, a segment is moving in the vertical plane to push the animal body upwards or downwards.

## 2.7. Analysing pelvic three-dimensional kinematics and its influence on hindlimb motion

To isolate and estimate pelvic rotation and observe its influence on the hindlimb, two models for each trial were created to examine the motion of the hindlimb in two alternative local anatomical reference frames: body axis and pelvic axis (figure 1f). Prior to generation of either model, the advancing motion of the body was removed by translating the X, Y, Z coordinates, creating a view as if our cameras were sliding down the trackway following the frog.

For the first model (fixed body axis model), kinematics data for each trial were fixed along the body axis which isolated pelvic lateral rotation, thus allowing us to measure pelvic angle. Specifically, the global translations and rotations of the frog body were mathematically fixed as if our camera were mounted to the frog spine itself.

For the second model, the fixed pelvic axis model, the kinematics were fixed along the pelvic axis, effectively removing any contribution of pelvic lateral rotation to hindlimb motion. To accomplish this, the lateral rotations of the pelvis were numerically removed as if the camera were fixed to the pelvis allowing observation of the motions of the leg and body from that fixed reference.

After transforming kinematics data for both models, we subtracted foot displacement in model 2 (fixed pelvic axis model) from that of model 1 (fixed body axis model) to quantify the contribution of the pelvis to stride length.

## 2.8. Generating model 1: the fixed body axis model

To create the fixed body axis model (Model FBA), the yaw motion of the body was fixed (equations (2.11)–(2.14)).

Although the frogs consistently walked in a straight line, their instantaneous body motions shifted medio-laterally in the dorsal view. Consequently, their trajectory was never precisely parallel with the trackway, causing variability in heading. The vector between the head and the vent markers represented the body axis vector ($\mathbf{v}_{Bod}$) and was used to define the x-axis of the body axis reference frame (figure 1f,g).

$$\mathbf{v}_{Bod} = P_{hea} - P_{ven} \tag{2.11}$$

The marker coordinates were fixed to the body axis by subtracting the body axis angle $\theta_{BA}$ (equation (2.12); figure 1f,g)

$$\theta_{BA} = \ acos\left(\frac{\mathbf{v}_{Bod} \cdot \mathbf{x}}{\|\mathbf{v}_{Bod}\|\|\mathbf{x}\|}\right) \tag{2.12}$$

where $\mathbf{x}$ represents the trackway X-axis [1, 0, 0] (equivalent to the X-axis in the global reference frame).

The X, Y, Z coordinates corresponded to a $12 \times 3$ matrix ($\mathbf{M}$, 12 markers each with X, Y and Z coordinates). The rotation matrix $\mathbf{R_1}$ (equation (2.13)) was used to rotate each marker, in each frame, about the Z-axis, by the angle $\theta_{BA}$, thus subtracting body axis motion (equation (2.14)). The resulting matrix ($\mathbf{M_{BA}}$) represented the body axis corrected marker coordinates for each given frame.

$$\mathbf{R_1} = \begin{bmatrix} \cos(\theta_{BA}) & -\sin(\theta_{BA}) & 0 \\ \sin(\theta_{BA}) & \cos(\theta_{BA}) & 0 \\ 0 & 0 & 1 \end{bmatrix} \tag{2.13}$$

$$\mathbf{M_{BA}} = [\mathbf{R_1} \ \mathbf{M}^T]^T \tag{2.14}$$

where $\mathbf{M}$ is a $12 \times 3$ matrix, $\mathbf{R_1}$ is a $3 \times 3$ matrix and $\mathbf{M_{BA}}$ is a $12 \times 3$ matrix.

The FBA model, created using the body axis corrected marker coordinates from $\mathbf{M_{BA}}$ then allowed motion of the hindlimb segments to be viewed from the perspective of the body axis reference frame. This essentially eliminated all lateral motions of the body with respect to the global reference frame and aligned direction of travel to the global X-axis while preserving all relative motion of the sacral and limb segments.

## 2.9. Generating model 2: the fixed pelvic axis model

For the second model, the fixed pelvic axis model (Model FPA), the yaw motion of the pelvis was fixed (equations (2.15)–(2.18)). A pelvic vector ($\mathbf{v}_{Pel}$) was computed using the body axis corrected marker coordinates and was used to define the pelvic axis reference frame (figure 1f). The $\mathbf{v}_{Pel}$ was used to calculate the fluctuation in pelvic vector angle ($\theta_{PV}$) throughout the stride cycle (equation (2.16)).

$$\mathbf{v}_{Pel} = P_{sac} - P_{ven} \tag{2.15}$$

where $P_{sac}$ and $P_{ven}$ are the marker coordinates from the FBA model.

$$\theta_{PV} = acos\left(\frac{\mathbf{v}_{Pel} \cdot \mathbf{x}}{\|\mathbf{v}_{Pel}\|\|\mathbf{x}\|}\right) \tag{2.16}$$

A clockwise rotation of $\mathbf{v}_{Pel}$ about the Z-axis was recorded as a negative angle, whereas an anti-clockwise rotation was recorded as a positive angle.

Marker coordinates were then fixed to the pelvic axis using rotation matrix $\mathbf{R_2}$ (equation (2.17)) to rotate matrix $\mathbf{M_{BA}}$ about the Z-axis by $\theta_{PV}$ (equation (2.18)) (figure 1h). The resulting matrix ($\mathbf{M_{PV}}$) represented the pelvic axis corrected marker coordinates for each frame.

$$\mathbf{R_2} = \begin{bmatrix} \cos(\theta_{PV}) & -\sin(\theta_{PV}) & 0 \\ \sin(\theta_{PV}) & \cos(\theta_{PV}) & 0 \\ 0 & 0 & 1 \end{bmatrix} \tag{2.17}$$

$$\mathbf{M_{PV}} = [\mathbf{R_2} \ \mathbf{M_{BA}}^T]^T \tag{2.18}$$

where $\mathbf{M_{BA}}$ is a $12 \times 3$ matrix, $\mathbf{R_2}$ is a $3 \times 3$ matrix and $\mathbf{M_{PV}}$ is a $12 \times 3$ matrix.

Model FPA preserved the motion of the hindlimb segments throughout the stride cycle and allowed us to observe the motion of the hindlimb without the contribution of pelvic motion by cancelling pelvic rotation.

## 2.10. Comparison between models 1 and 2

To investigate the potential impact of $v_{Pel}$ rotation on hindlimb stride length, the two models were compared. The cranio-caudal excursions of the hindlimb joints (TMT, ankle and knee) in the FPA model were compared with those in the FBA model during a single stride cycle (stance to stance). Cranio-caudal excursion was calculated by taking the Euclidean distance between each joint position at the onset of stance phase and the frame of maximum hip extension (i.e. maximum limb retraction) and was used as a proxy for stride length (forward propulsion of the animal's body). The difference in the cranio-caudal hindlimb joint excursions between the two models was referred to as the hypothetical gain in stride length and was calculated by subtracting the joint excursions in Model FPA from those in Model FBA.

## 2.11. Statistical analysis

Since pelvic oscillation is sinusoidal in salamanders [29], we expected a sine wave pattern from pelvic lateral rotation in frogs. A test of significance was applied to the $\theta_{PV}$ data for each trial to discern pelvic lateral rotation from random fluctuations in $\theta_{PV}$ due to digitizing noise. This was achieved using custom-written code to test the significance of the signal over digitizing noise [36,37]. A $p$-value of less than 0.05 indicated that a wave pattern was present in the data and not an artefact of digitizing noise.

Consistent with prior work on frog kinematics (e.g. [38]), we pooled data from all individuals to illustrate how kinematic traces from individual trials compare with means pooled over all individuals. Traces averaged by each individual are shown in the electronic supplementary material, figure S1. Additionally, we used linear mixed-effects models (LMMs) to discern the influence of individual frogs on the trends observed using the LMM package in R (R Core Team, 2013, Vienna) via nlme::lme. LMMs were computed with stride speed and pelvic angle, hypothetical gain in joint excursion (at the TMT, ankle and knee joint) and pelvic angle as fixed effects, and individual frog as a random effect. LMMs were additionally computed with stride frequency, length and speed as fixed effects, and individual frog as a random effect. $\Omega^2$ values (analogous with $R^2$) were computed as a measure of explained variation by the LMMs [39].

# 3. Results

## 3.1. Animal morphology

A full overview of the morphological characteristics of each individual frog is provided in table 1. Mean frog body width was $30.26 \pm 6.09$ mm, while measurements of hindlimb segment lengths were $16.43 \pm 2.66$ mm hip-to-knee, $19.05 \pm 2.28$ mm knee-to-ankle, $10.95 \pm 1.22$ mm ankle-to-TMT and $22.01 \pm 2.60$ mm TMT-to-digit. All data are presented as mean $\pm$ standard deviation (s.d.).

## 3.2. Footfall patterns

All frogs used alternating contralateral forelimb and hindlimb pairs in a lateral sequence walk (figure 2). While at least two limbs were in contact with the ground at all times, the forelimbs entered swing phase and stance phase slightly before the contralateral hindlimb, ensuring at least three limbs maintained ground contact throughout the majority of the stride cycle. Despite travelling over a range of speeds from 0.04 to 0.23 m s$^{-1}$ (mean $0.14 \pm 0.05$ m s$^{-1}$), all frogs used a duty factor consistently greater than 0.50, indicative of walking gait [4].

## 3.3. Descriptive hindlimb joint kinematics

Hindlimb joint motions were described using three-dimensional vector angles between the limb segments corresponding to the hip, knee and ankle joints ([16]; see Material and methods). Observation of the segments both in the raw video data and the resulting kinematic models suggested that hindlimb motion throughout the stride cycle is complex, with limb segments not restricted to a single plane. In addition to protraction and retraction in the frontal plane, rotation of the skin markers and limb surfaces suggested movement in multiple planes throughout the stance–swing cycle. Three general patterns emerged among the intricate limb segment motions: (i) the hip, knee and ankle joints

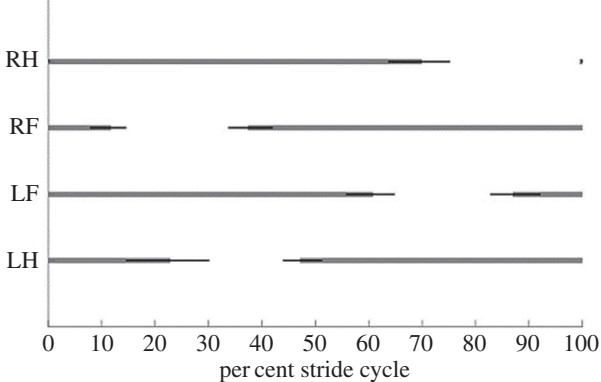

**Figure 2.** Mean footfall pattern (grey bars) $\pm$ 1 s.d. (black bars) obtained from pooled data from all trials (8 frogs, 23 trials total). RH, right hindlimb; RF, right forelimb; LF, left forelimb; LH, left hindlimb.

protracted and retracted cranio-caudally, rhythmically and in-phase, creating sinusoidal curves (figure $3a-c$), (ii) a relatively longer stance phase compared with swing phase introduced asymmetry to the curves, and (iii) although joint retraction occurred simultaneously with the onset of stance phase (0% stride cycle), maximum joint angle did not occur simultaneously with the onset of swing phase, since limb protraction was initiated before the foot broke contact with the ground.

The polar coordinates of joint orientation demonstrated that all limb segments had a larger lateral range of motion compared with longitudinal range of motion (figure $3d-i$). This finding suggests the majority of hindlimb motion occurs as cranio-caudal protraction and retraction in the frontal (horizontal) plane. The relatively small range of dorsoventral longitudinal motion in the vertical plane suggests vertical motion of the limb segments contributes only modestly to overall limb segment motion. Compared with the femur (proximal) and the tarsals (distal), the tibiofibula shows little range of motion in either plane, implying that the tibiofibula maintains a similar orientation throughout the stride cycle.

## 3.4. Pelvic vector kinematics and estimated joint excursion as a function of pelvic vector rotation

The angular excursion of $\mathbf{v}_{Pel}$ ($\theta_{PV}$) fluctuated throughout the stride cycle, creating triangular waves. Although the extent of lateral rotation estimated by Model FBA was variable among trials, all traces were statistically significant ($p \ll 0.001$) as per Fisher's exact $g$-test for time-series data [36,37]. Among pooled data, $\mathbf{v}_{Pel}$ underwent a mean angular excursion of $12.71 \pm 4.40°$ (mean $\pm$ s.d.) with respect to the body axis (figure $4a$ and table 2). Mean $\mathbf{v}_{Pel}$ lateral rotation occurred in phase with the mean hindlimb three-dimensional joint angles (peaking at approx. 60% of the stride cycle). The $\theta_{PV}$ angle values increased during hindlimb joint retraction (extension) and began to decrease during limb protraction (flexion). Again, peak $\theta_{PV}$ did not occur simultaneously with the onset of swing phase (figure $4a$).

Model FPA allowed hindlimb motion to be estimated without the contribution of the pelvis. The estimated influence of pelvic lateral rotation on the hindlimb was therefore quantified by comparing the cranio-caudal excursion of each hindlimb joint between the two models and calculating the hypothetical gain (see Material and methods). The cranio-caudal excursions of all three right hindlimb joints in one stride were consistently less in Model FPV joints compared with Model FBA, as indicated by the positive hypothetical gain values (table 2 and figure $4b,c$). Positive hypothetical gain values show that lateral rotation of $\mathbf{v}_{Pel}$ increased the cranio-caudal excursions of the knee, ankle and TMT. Hypothetical stride length gain values were always largest at the distal-most joint, the TMT.

## 3.5. Relationship between pelvic vector rotation, hindlimb motion and speed

Pelvic lateral rotation was positively correlated with the hypothetical gain in cranio-caudal excursion of all three hindlimb joints ($p < 0.03$ for hip, knee and TMT). The larger the angular excursion of the pelvis the greater the contribution to cranio-caudal hindlimb excursion

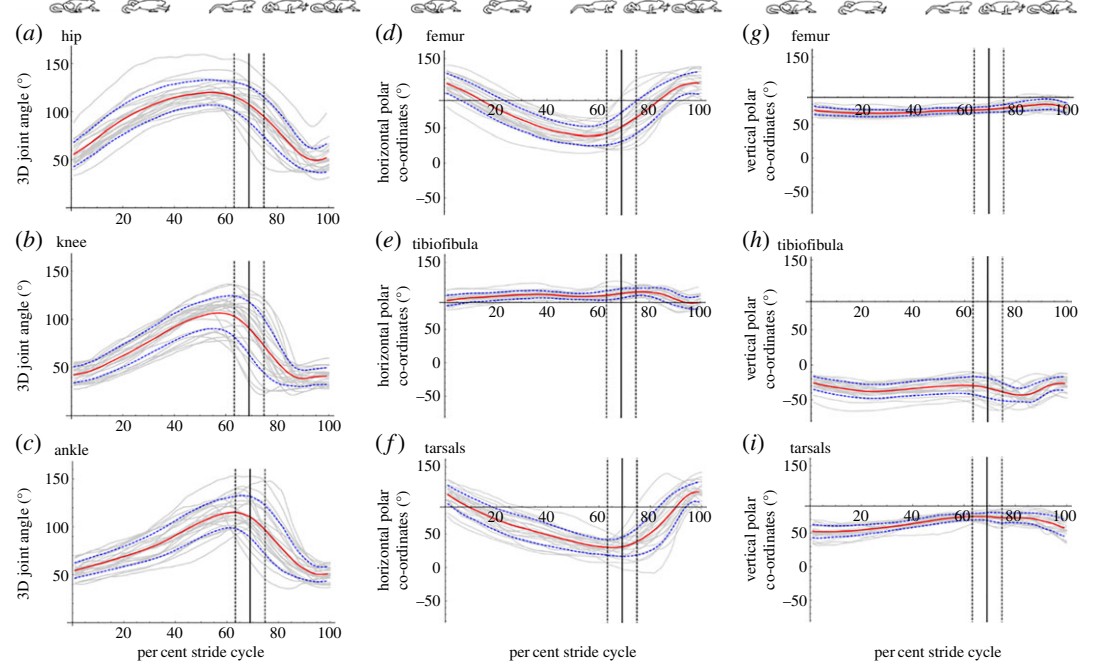

**Figure 3.** Mean three-dimensional joint angles for hip (*a*), knee (*b*) and ankle (*c*) joints (red lines) $\pm 1$ s.d. (blue dashed lines) and mean polar coordinates to describe femur (*d* and *g*), tibiofibula (*e* and *h*) and tarsal (*f* and *i*) motion horizontally (*d*–*f*) and vertically (*g*–*i*) (red lines) $\pm 1$ s.d. (blue dashed lines). The onset of stance occurs at 0% stride cycle and mean onset of swing phase is denoted by the solid black line $\pm 1$ s.d. (black dashed lines). Data obtained from pooled data from all trials (8 frogs, 23 trials total). Individual trials shown in grey. For polar coordinates plots (*d*–*i*), horizontal motion occurred in the frontal plane and vertical motion occurred in the transverse plane.

(figure 5*a*–*c*; electronic supplementary material, table S1). However, lateral rotation showed a negative relationship with speed, whereby as the animals travelled faster they rotated their pelves to a lesser degree (figure 6*b*).

## 3.6. Spatio-temporal kinematic data

*Kassina maculata* favoured an increase in stride frequency, as opposed to stride length, as they increased speed (figure 6*a*). The range of speeds at which the animals travelled increased approximately sixfold from 0.04 to 0.23 m s$^{-1}$. Whereas stride frequency also increased approximately sixfold across the range of speeds travelled ($\Omega^2 = 0.972$; $p < 0.001$), there was no relationship between stride length and speed ($\Omega^2 = 0.463$; $p = 0.489$; figure 6*b*; electronic supplementary material, table S1).

# 4. Discussion

We aimed to quantify pelvic lateral rotation and evaluate its impact on hindlimb stride length during walking in *Kassina maculata*. Building on prior work which established that frogs use well-developed musculature to actively rotate the pelvis [6], we further show that pelvic oscillations are consistent and spatio-temporally coordinated with the leg in a manner favourable for augmenting stride length. The specific motions are as follows; at the beginning of stance for the leading (right) hindlimb, the pelvis is at its most 'flexed' position, oriented towards the leading side (i.e. the base where the ilia connect points to the right side). As the right leg extends caudally, the pelvis rotates synchronously towards the opposite (left) side, helping to translate the leg farther backwards (figure 4*a,b*). In spite of the tight spatio-temporal coordination, our results demonstrated that the cranio-caudal excursion of the hindlimb only modestly increases with pelvic lateral rotation. While our hypothesis is technically supported, pelvic rotation can impact stride length, the increase is minimal and it therefore seems unlikely these animals rely on pelvic lateral rotation as a stride length augmenting mechanism.

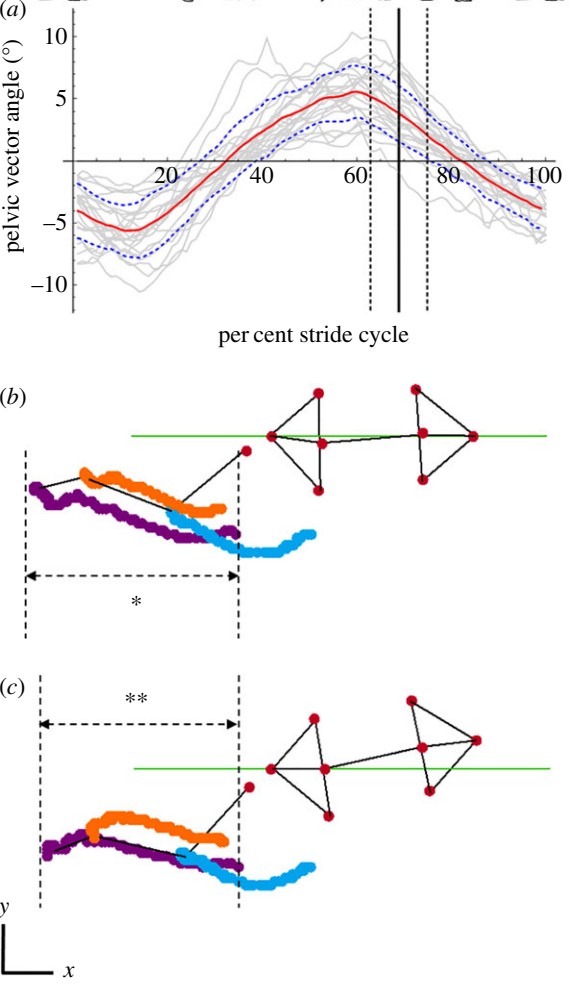

**Figure 4.** Mean pelvic vector angles ($\theta_{PV}$) (red line) $\pm 1$ s.d. (blue dashed lines) (*a*). The mean onset of swing phase in the $\theta_{PV}$ plot is denoted by the solid black line $\pm 1$ s.d. (black dashed lines), obtained from pooled data from all trials (8 frogs, 23 trials total). Individual trials shown in grey. An example trial showing cranio-caudal hindlimb joint excursions at maximum hip extension from both Model FBA (*b*) and Model FPA (*c*) for comparison. Colours denote: TMT, purple; ankle, orange; knee, blue. Green line denotes X-axis. Vertical dashed lines represent joint excursion boundaries for the TMT joint in each model; the horizontal dashed arrows provide an example of cranio-caudal joint excursion for the TMT joint in each model. The hypothetical gain value is the difference between distance * and distance **.

## 4.1. The degree of pelvic lateral rotation is variable among frog species and within individuals

Given that the pelvis oscillates sinusoidally in salamanders [29], we predicted the same to be true of the pelvic lateral rotation in frogs. Although the angular excursion of $\mathbf{v}_{Pel}$ indeed fluctuated throughout the stride cycle in a wave pattern, the wave form generated was more triangular shaped than seen in salamanders. Since we digitized one full stride cycle, we make the assumption that this pattern repeats as a waveform across multiple strides. We are confident we are safe in this assumption as we ensured we were using a representative 'middle stride' where the animal was moving at a steady speed in a straight line (see Material and methods).

Throughout a single hindlimb stride cycle in *K. maculata*, the pelvis rotated through an arc of approximately $13°$ on average ($\pm 6.5°$ either side of the body axis). This value falls within the range of lateral rotation of the pelvic girdle during walking presented by Emerson [32] for other walking frog species: *Bufo boreas* ($N = 5$, mean $8.8°$), *Phyrnohyas venulosa* ($N = 5$, mean $3.6°$) and *Rhinophrynus dorsalis* ($N = 6$, mean $14.2°$), although it is not clear whether angles reported by Emerson are total bilateral excursion about the midline or rotation to one side. Nonetheless, the degree of pelvic rotation varies between the above species measured despite all presumably having the same pelvic articulation type. *Phyrnohyas venulosa*, for example, had noticeably lower reported angular excursions [32]. This

**Table 2.** Kinematic data obtained from FBA and FPV Models. Mean values presented as mean $\pm$ s.d. Hypothetical gain refers to the difference in cranio-caudal excursion between the two models.

| animal number | trial number | speed (m s$^{-1}$) | $v_{Pel}$ total angular excursion (degrees) | hypothetical gain in cranio-caudal excursion (mm) | | |
|---|---|---|---|---|---|---|
| | | | | TMT | knee | ankle |
| 1 | 1 | 0.04 | 16.16 | 4.54 | 3.58 | 3.80 |
| 2 | 1 | 0.09 | 17.58 | 4.57 | 3.64 | 3.17 |
| | 2 | 0.06 | 13.53 | 4.26 | 3.34 | 3.26 |
| | 3 | 0.09 | 14.71 | 4.70 | 3.43 | 3.35 |
| 3 | 1 | 0.13 | 20.94 | 7.28 | 3.92 | 2.74 |
| | 2 | 0.10 | 16.55 | 3.88 | 3.92 | 2.74 |
| | 3 | 0.12 | 15.68 | 5.68 | 4.88 | 3.77 |
| 4 | 1 | 0.11 | 14.27 | 6.91 | 5.50 | 5.62 |
| | 2 | 0.13 | 11.30 | 5.09 | 4.50 | 3.84 |
| 5 | 1 | 0.10 | 13.28 | 3.86 | 3.03 | 2.98 |
| | 2 | 0.18 | 8.63 | 2.05 | 1.65 | 1.68 |
| | 3 | 0.14 | 11.38 | 2.11 | 1.64 | 1.60 |
| | 4 | 0.12 | 11.90 | 3.13 | 2.39 | 2.51 |
| | 5 | 0.11 | 12.75 | 5.48 | 3.89 | 4.18 |
| 6 | 1 | 0.19 | 5.78 | 1.28 | 0.99 | 0.90 |
| | 2 | 0.23 | 5.04 | 1.34 | 1.01 | 1.06 |
| | 3 | 0.22 | 4.30 | 0.89 | 0.66 | 0.62 |
| 7 | 1 | 0.16 | 14.33 | 2.63 | 2.39 | 2.10 |
| | 2 | 0.09 | 16.18 | 3.34 | 2.76 | 2.53 |
| 8 | 1 | 0.23 | 7.19 | 2.00 | 2.01 | 1.23 |
| | 2 | 0.16 | 18.13 | 4.47 | 4.56 | 2.46 |
| | 3 | 0.20 | 13.93 | 4.01 | 3.48 | 2.65 |
| | 4 | 0.17 | 8.65 | 2.05 | 1.73 | 1.46 |
| mean | | 0.14 $\pm$ 0.05 | 12.71 $\pm$ 4.40 | 3.72 $\pm$ 1.75 | 3.00 $\pm$ 1.33 | 2.62 $\pm$ 1.20 |

may have been because measurements for this species were obtained from vertical walking (i.e. climbing) sequences; it is possible that vertical walking kinematics differ compared to walking on a horizontal surface. On the other hand, morphological variation could be an impacting factor. We now know that different forelimb : hindlimb proportions of different species demand different kinematics [5], and thus it is likely that pelvic length and width ratios could also result in altered hindlimb kinematics. The effect of pelvic morphological variation on hindlimb and pelvic kinematics is something that should be considered in future comparative kinematics studies.

In the data presented here for *K. maculata*, the degree of lateral rotation also varies with speed of travel, suggesting the degree of lateral rotation exhibited by frogs during walking is variable depending on species, terrain, gradient and speed. In future work, this hypothesis could be addressed by applying our kinematic analyses to study additional walking species across a range of terrains at a variety of speeds.

## 4.2. Comparisons with other sprawling tetrapods

Pelvic motion also contributes to hindlimb excursion in other sprawling tetrapods. Cryptodire turtles have a mobile sliding joint connecting the ilium and sacrum, permitting pelvic yaw during locomotion [40]. These turtles exhibited a slightly higher mean pelvic angular excursion than observed

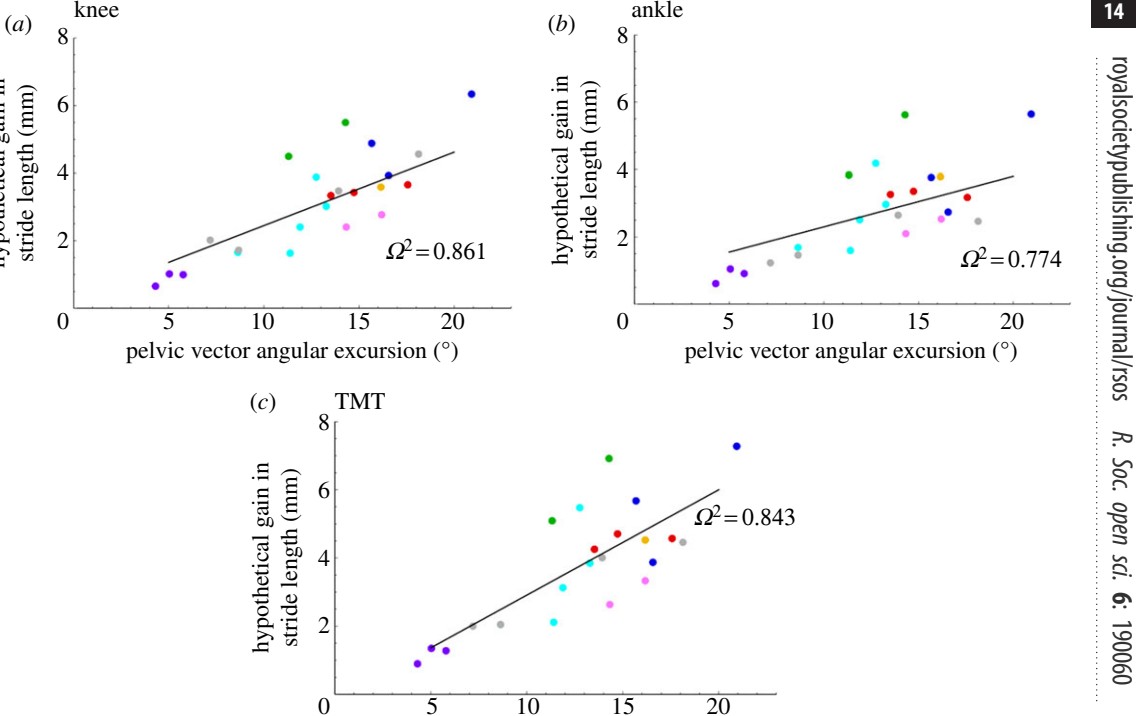

**Figure 5.** Linear mixed model showing the relationship between the hypothetical gain in cranio-caudal excursion for the right hindlimb joints and the total angular excursion of the pelvic vector ($\mathbf{v}_{Pel}$). (a) knee, (b) ankle, (c) TMT. Animal 1, yellow marker; Animal 2, red marker; Animal 3, blue markers; Animal 4, green markers; Animal 5, cyan markers; Animal 6, purple markers; Animal 7, pink markers; Animal 8, grey markers. $\Omega^2$ computed as 1-(var(residuals(model))/(var(getResponse((model))))) using nlme::lme.

in frogs (18.2°); however, they are an order of magnitude larger in body size and have a shorter relative limb length. Increased degree of pelvic rotation was found to correlate with increased femoral excursion, as was the case in the present study [40].

Salamanders, lizards and alligators also use pelvic rotation during walking ([29,41,42], respectively); however, rotation of the pelvic girdle in these animals is less comparable to that of the frogs due to anatomical differences in pelvic girdle structure. In the Pacific giant salamander (*Dicamptodon tenebrosus*), Ashley-Ross measured mean anterior–posterior trunk angle ranges of 65.7° and 88.1° during walking and trotting, respectively [29]. These angles account for motion of both the anterior and posterior spine, whereas the data presented here for *K. maculata* only accounts for motion of the pelvis (posterior spine in this comparison). The degree of angular excursion seen in *K. maculata* is therefore conservative compared with data presented by Ashley-Ross for salamanders.

## 4.3. Pelvic lateral rotation only modestly increases stride length and decreases with speed

Pelvic lateral rotation, whether generated through axial bending of the spine (as in other sprawling tetrapods) or pelvic lateral rotation, does not guarantee increased stride length. In order for the rotation of the axial components of the body to contribute to the cranio-caudal excursions of the leg, both spatial alignment and temporal synchronization between leg segments and pelvis are required. It is only in instances where lateral rotation is spatio-temporally coordinated with limb segment protraction/retraction in the frontal plane that cranio-caudal excursion is increased. Providing that the distal limb joint motions are synchronized with the cranio-caudal component of the proximal limb segment (but not the dorsoventral component), the impact of lateral rotation of the pelvis is cumulative proximo-distally, amplifying distal limb segment motion. For the above reasons, a measurement of pelvic lateral displacement (e.g. [32]) in isolation of three-dimensional limb kinematics is not sufficient to determine the pelvis' impact on walking.

In the kinematic data presented here, both conditions (as explained above) were satisfied. The majority of hindlimb motion was protraction and retraction in the frontal plane and this hindlimb motion was

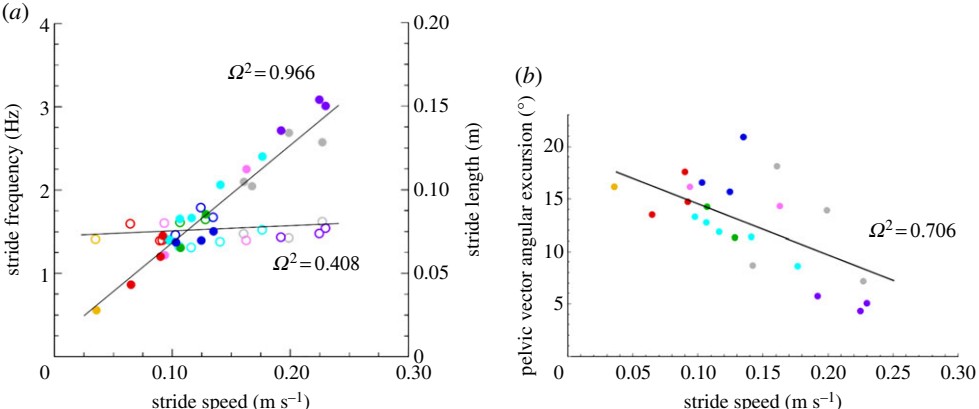

**Figure 6.** Linear mixed model showing relationship between stride speed (m s$^{-1}$) and: stride length (open symbols; *a*); stride frequency (filled symbols; *a*); and the total angular excursion of the pelvic vector ($\mathbf{v}_{Pel}$) (*b*). Animal 1, yellow marker; Animal 2, red marker; Animal 3, blue markers; Animal 4, green markers; Animal 5, cyan markers; Animal 6, purple markers; Animal 7, pink markers; Animal 8, grey markers. $\Omega^2$ computed as 1-(var(residuals(model))/(var(getResponse((model))))) using nlme::lme.

temporally synchronized with pelvic lateral rotation. Consequently, the spatio-temporal conditions were sufficient for pelvic lateral rotation to influence hindlimb motion. Lateral rotation of the pelvis in *K. maculata* produced modest, but measurable, increases in the cranio-caudal excursion of the hindlimb joints (used here as a proxy for stride length). However, with pelvic excursion translating, on average, to 3.72 mm hypothetical increase in stride length at the TMT (table 2), it is unlikely these frogs experience any benefit from such a modest increase. Furthermore, increased speed of travel was coupled with a decrease in pelvic lateral rotation while stride length remained unchanged; instead, greater speeds were driven by increased stride frequency. Pelvic lateral rotation does not appear to be a mechanism for increasing speed.

These findings are contrary to other species; Mayerl *et al*. demonstrated that in both walking and swimming cryptodire turtles (*Pseudemys concinna*) lateral rotation of the pelvis increased as the animals increased their speed [40]. Similarly, salamanders increase their trunk angular excursion when increasing their speed [29]. Increased lateral rotation at increased speeds may not be a viable option for frogs. The centre of mass (COM) in anurans is positioned just dorsal to the sacrum [38]. Since the ilio-sacral joint is the joint about which the ilia rotate, pelvic lateral rotation acts to shift the COM laterally throughout a stride. Increasing speed by means of increasing the degree of lateral rotation and subsequently stride length would result in both larger and faster COM displacements. It is probable that such displacements are detrimental to the animal's stability, leaving it unable to counterbalance the angular momentum created by rapidly shifting and swinging a large proportion of their mass. Furthermore, walking frogs may suffer a constraint due to their long hindlimbs relative to their forelimbs. The forelimb joints already reach full extension during the stride cycle suggesting that any significant further increase in hindlimb stride would be impossible to maintain, and as such walking frogs simply may not be able to benefit from increased stride length [5].

## 4.4. *Kassina maculata* show no evidence for kinematically distinct gaits during quadrupedal locomotion

Despite a sixfold increase in speed, little variation in footfall pattern was observed (figure 2; [5, fig. 5]). Other sprawling tetrapods, however, change gaits as they travel faster. For example, lizards and salamanders switch to a trotting gait [29,41,43] at higher speeds. Alligators maintain a walking-trot gait across a twofold increase in speed, although they typically transition to a gallop at high speeds [42]. By contrast, there exists no strong evidence that anurans switch from walking to another distinct quadrupedal gait (e.g. trotting). Although there is some evidence that COM kinematics may shift in a manner consistent with bouncing gaits [4], walking frog species maintain remarkably consistent hindlimb kinematics across speed (figure 3; [5, fig. 8]). Rather than shifting quadrupedal gaits, other frog species (e.g. *Bufo woodhousii fowleri*) are known to increase their speed via a transition to hopping [7,8]. *Kassina maculata*, however, is not known to switch gaits at all, opting to maintain their

quadrupedal gait across a wide range of speeds without switching to hopping [4,5]. Thus, while *K. maculata* shares features of walking with the other sprawling tetrapods, in particular showing remarkably similar walking footfall patterns to the giant coastal salamander [4], this frog maintains a lateral sequence walking gait at all speeds, consistent with prior studies [4,5].

Using non-invasive methods, this study was able to quantitatively show that while pelvic lateral rotation in *K. maculata* has the potential to increase hindlimb stride length, the benefits appear modest. It should be noted that the methods used in this study cannot provide data on internal skeletal movements or visualize long-axis segment rotations. Furthermore, the external markers placed on the joints represented the best estimate of the joint centres of rotation. Consequently, data presented in this study, while quantitative, are estimates based on the gross movements of the hindlimb and body. The use of non-marker-based fluoroscopy [44] in further experiments could be used to verify pelvic motion. Furthermore, X-ray of moving morphology (XROMM) experiments would allow precise pelvic morphology to be observed and therefore help resolve whether pelvic morphology and spine flexibility might constrain the angular movements of the ilia.

Additionally, Model FPA cannot be used as a representation of walking kinematics in the absence of pelvic lateral rotation. It is likely that, were frogs to have this motion fixed anatomically, they would display compensatory motions which could have an effect on stride length. While these models cannot predict hindlimb motions with a fixed pelvis, or subsequent compensatory behaviour, they do demonstrate that pelvic lateral rotation has the potential to modestly increase hindlimb stride length.

## 5. Conclusion

Lateral rotation has been shown to augment walking/running stride length in other tetrapods. While we show that pelvic lateral rotation has the potential to increase stride length very modestly, our evidence suggests that the mobile pelvis in frogs is unlikely to be a walking performance-enhancing feature. In the light of the great diversity in pelvic morphology within pelvic types [1] it remains unclear how species-specific skeletal morphology influences locomotor behaviour. Using the methods of the present study as a template, future kinematics studies should be conducted across a wide sample of frog taxa to establish whether other walking species employ similar limb-pelvis mechanics. Our current work, along with the suggested future studies, will help clarify broad relationships between musculoskeletal diversity and locomotor performance crucial for understanding the evolution and radiation of frogs and other tetrapods.

Ethics. All procedures were approved under the Home Office Licence no. 70/8242.

Data accessibility. All custom code (MATHEMATICA and R notebook scripts) written for and used in this study are available on Github: https://github.com/frogtronics/pelvicKinematics.

Authors' contributions. Design and construction of experimental set-up: A.J.C., L.B.P. and C.T.R.; data acquisition, read and revised manuscript: A.J.C., L.B.P., C.H. and C.T.R.; data analysis and interpretation: A.J.C., C.H. and C.T.R.; wrote MATHEMATICA code and drafted the manuscript: A.J.C. and C.T.R. All authors provided final approval of the submitted version to be published.

Competing interests. There were no competing interests in the current study.

Funding. Funding for this work was provided by a European Research Council Starting grant no. (PIPA338271) awarded to C.T.R.

Acknowledgements. Many thanks go to Alastair Wallis, Dale McCarthy, Michael Murphy and David Lathlean for the care of the research animals. Additionally, thanks go to Ludovic Pelligrand and Chris Handley for their assistance with veterinary care and anaesthetic dosing, to Timothy West for providing software access, and to Hannah Safi and Charlotte Nwanodi who assisted in the collection of high-speed video data. Authors would also like to thank Sandy Kawano and Simon Wilshin for their assistance in 'R', as well as James Charles for proof-reading draft versions of the manuscript. Finally, authors would like to thank the editor and reviewers for their time and valuable comments.

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
