## [Reviewer comments · Royal Society Open Science]

Review History

RSOS-190060.R0 (Original submission)

Review form: Reviewer 1

Is the manuscript scientifically sound in its present form?

Yes

Are the interpretations and conclusions justified by the results?

Yes

Is the language acceptable?

Yes

Is it clear how to access all supporting data?

Yes

Do you have any ethical concerns with this paper?

No

Have you any concerns about statistical analyses in this paper?

No

Recommendation?

Accept with minor revision (please list in comments)

Comments to the Author(s)

Dear Editor,

I have considered the manuscript by Collings et al. on 'The impact of pelvic lateral rotation on hindlimb kinematics and stride length in the red-legged running frog, *Kassina maculata*'. Generally spoken, I like this contribution and I have only some minor comments, suggestions and questions.

Line 85: What is meant with asynchronous (explain)?

Line 90: I think there does exist (yet probably not too detailed) kinematics of crawling (walking) in frogs (e.g. Nauwelaerts & Aerts, 1992; J. Zool.).

Line 134: This is a rather arbitrary definition of transition between swing and stance and vice versa, which is fine as such, but probably more prone to errors in its determination than the rigorous and most often used 'first contact' and 'last contact' of the hands and feet. Maybe give a short argumentation why this definition is used.

Line 173: Define 'body side' (as a segment) or at least refer explicitly to figure 1b.

Line 175: I find the use of 'V' to indicate (some) vectors rather confusing (at first I thought this referred to speed ('V'elocity). Why not using conventional vector script? (see also further).

Line 197: New notations are used for the same vectors.

Line 222: Increase length of the square root symbol (must encompass Z^2).

Line 223&226: I suggest to use another variable name. Presently flexion/extension angles are identical to joint angles (line: 197)

Line 251: This must be 'contribution of the pelvis' (in model 2 the hip also contributes because of femur rotation)

Line 251-258: Personally, I do not think this explanation is really needed.

Line 277: Judging what follows, M must be replaced by MT.

Line 282: use matrix notation and remove capitals in Cos and Sin

Line 284: remove '.' between R1 and MT. Moreover, I wonder whether the notation of equation 14 is correct (most likely calculations are OK). At first glance, R1 must not be transposed. Furthermore, notice that according this equation, MBA becomes a 3x12 matrix. As such, the explanation in lines 306-312 becomes very confusing (rotation of MBA). This means that MBA must not be transposed for the rotation, although this is done according to equation 18. On the other hand, this equation refers to MBAT, a matrix that is further not defined but probably means 'the transposition of MBA'. As said, this is all very confusing and in general I think it will be

beneficial of the authors consult a mathematician to check and clean up the mathematical script (including the vector notations).

Line 385: sounds a bit odd. What is exactly meant here?

Line 410: $0.04\text{m/s} \Rightarrow 0.26\text{m/s}$ is a 6fold increase (not 3fold) and seems to accord to what can be seen in figure 6A

Line 411: If stride length is not changing, frequency must increase with a factor similar to that of speed (i.e. 6-fold; cf. above). This is confirmed by the frequency range in figure 6A ($0.5\text{hz} \Rightarrow 3\text{hz}$). Hence, the 3fold and 10fold increases mentioned in this part are confusing/puzzling.

Line 508: 6fold? (cf.above).

Review form: Reviewer 2

Is the manuscript scientifically sound in its present form?

Yes

Are the interpretations and conclusions justified by the results?

Yes

Is the language acceptable?

Yes

Is it clear how to access all supporting data?

Yes

Do you have any ethical concerns with this paper?

No

Have you any concerns about statistical analyses in this paper?

No

Recommendation?

Major revision is needed (please make suggestions in comments)

Comments to the Author(s)

Review: The impact of pelvic lateral rotation on hindlimb kinematics and stride length in the red-legged running frog, *Kassina maculata*.

This study uses a clever non-invasive computational method to quantify the contribution of lateral pelvic movements to the stride length of walking frogs. The analysis is non-trivial yet done carefully and explained well and succinctly. Pelvic anatomy and function in frogs is a research topic that remains largely unexplored and this study will be a significant contribution. Overall, I really enjoyed the manuscript and would like to see it published. However, I would like to suggest some edits that I think would greatly improve the impact of this study.

First, I was hoping for a maximum calculated stride length increase with each degree of pelvic rotation, based purely on the geometry of the skeleton (Fig. 1B). Given how short and rigid the spine of the frog is I would not expect there to be much bending near the pelvis. Furthermore, the

pelvis itself is narrow and so even large angular movements of the could conceivably have a small effect on stride length. So I was hoping for a hypothetical angle calculate, in the best case scenario, i.e. all joint extensions occur synchronously so that proximal rotation is fully transmitted to the distal limb.

The other thing I expected to see in the discussion was a constraint on stride length posed by the shorter forelimbs. For example, if the forelimb joints are near full extension during the stride an increase in hindlimb stride length would not be possible because the forelimbs would not be able to keep up.

Secondly, it was unclear for much of the manuscript what the two models were that the authors compared to each other. This was first described on line 238, even though there were several allusions to two approaches earlier (E.g. line 88, 162). I would have liked a brief description even in the abstract. At the end of the introduction (line 87) the first model is briefly described but the comparison remained incomplete.

Specific remarks

Line 22: IS - I would spell out this anatomical structure for the sake of those who are not frog anatomy fanatics (I don't know who these would be, but one never knows...!)

Line 24: range of speeds - units missing

Line 27: The reporting of results could use some more clarification. The last three sentences of the summary only made sense after reading the whole manuscript.

Line 68: "From salamander studies" could be omitted.

Line 70-71: I could find the illustration of this point in Figure 1.

Line 73-80: I had trouble seeing how pelvic rotation could not contribute to stride length increase.

Line 77-78: is this muscle activation pattern (citation 6) measured during walking or frog jumping?

Line 87: "The first model..." there is no follow up with the second model.

Line 162: "... using two methods" this made we wonder whether these were the two models being compared.

Line 176: "in terms of the skin marker landmarks". I expected some guidance on where to find the names for the landmarks that are given in the next page. It wasn't obvious what "P" in Pkne stood for...

Line 229-235: Excellent paragraph explaining the benefits of using polar coordinates!

Line 237-240: This was the first description of the 2 models. I would have liked to see it much earlier in the manuscript. The authors do a good job of explaining the two models succinctly so it should not be a problem to introduce them earlier.

Line 251-258: I thought the authors did a terrific job explaining global vs local coordinate systems before line 251. The further explanation with a different example (throwing a ball) could be omitted.

Line 329-330: I would have liked a little more explanation of why a sinusoidal motion in PV was expected. Is that due to the periodicity of the measurements? How can that be achieved if only one stride per sequence is digitized? I think this is a valuable contribution of this manuscript and strengthens the dataset despite the relatively small number of strides digitized. It is also not a frequently done analysis, even though it certainly should be. In my opinion, it therefore warrants a little more description.

Line 357: Is the upper limit of the speed observed in this study (0.23 m s⁻¹) close to the maximum for this species? At what speed does this frog species switch to hopping? This is perhaps a point for the discussion.

Line 368: "...creating sinusoidal curves" these are not the same sinusoidal waves as expected in the tests for digitizing error, correct?

Line 377: "...contributed only modestly to limb segment motion". This result was not obvious to me from the reference to figure 3D-I in line 375.

Line 404-406: I did not understand the difference between V_{pel} and lateral rotation, so I could not see how they had opposite relationship with speed (Fig. 5 vs 6).

DISCUSSION

Overall I had a hard time following the discussion to the final conclusion. There were often large jumps from some exceptional descriptions of complex 3-dimensional kinematics (e.g. line 467-478) to conclusions that did not follow directly.

Line 430-448: This is where I expected a discussion about the differences in pelvis length or structure among the different species. A similar anatomical comparison would be very useful in the following section on other sprawling tetrapods, as well.

Line 491: "... to 3.78 mm hypothetical increase in stride length" what is the origin of this hypothetical value?

Line 499-505: Great explanation and discussion of the lateral COM movements and the potential contribution to instability.

Line 517-520: In this section I expected discussion of the gait transition in *Kassina*, from walking to jumping.

Line 533-535: This sentence seemed contradictory within itself.

Decision letter (RSOS-190060.R0)

12-Mar-2019

Dear Ms Collings

On behalf of the Editors, I am pleased to inform you that your Manuscript RSOS-190060 entitled "The impact of pelvic lateral rotation on hindlimb kinematics and stride length in the red-legged running frog, *Kassina maculata*" has been accepted for publication in Royal Society Open Science

subject to minor revision in accordance with the referee suggestions. Please find the referees' comments at the end of this email.

The reviewers and handling editors have recommended publication, but also suggest some minor revisions to your manuscript. Therefore, I invite you to respond to the comments and revise your manuscript.

- Ethics statement

- Data accessibility

If you wish to submit your supporting data or code to Dryad (<http://datadryad.org/>), or modify your current submission to dryad, please use the following link:
<http://datadryad.org/submit?journalID=RSOS&manu=RSOS-190060>

- Competing interests

- Authors' contributions

- Acknowledgements

- Funding statement

Because the schedule for publication is very tight, it is a condition of publication that you submit the revised version of your manuscript before 21-Mar-2019. Please note that the revision deadline will expire at 00.00am on this date. If you do not think you will be able to meet this date please let me know immediately.

Supplementary files will be published alongside the paper on the journal website and posted on the online figshare repository (<https://rs.figshare.com/>). The heading and legend provided for each supplementary file during the submission process will be used to create the figshare page,

so please ensure these are accurate and informative so that your files can be found in searches. Files on figshare will be made available approximately one week before the accompanying article so that the supplementary material can be attributed a unique DOI.

on behalf of Professor Emily Standen (Associate Editor) and Kevin Padian (Subject Editor)
openscience@royalsociety.org

Associate Editor Comments to Author (Professor Emily Standen):

Dear Dr. Collings,

Thank you for the submission of your paper entitled "The impact of pelvic lateral rotation on hind limb kinematics and stride length in the red-legged running frog, *Kassina maculata*. This is a very nicely written paper and an interesting approach.

The reviews are generally very positive about this paper but I feel as though they do have some comments that are worth considering. I look forward to seeing your revised manuscript.

Sincerely,
Emily Standen

Reviewer comments to Author:
Reviewer: 1

Comments to the Author(s)
Dear Editor,

I have considered the manuscript by Collings et al. on "The impact of pelvic lateral rotation on

hindlimb kinematics and stride length in the red-legged running frog, *Kassina maculata*. Generally spoken, I like this contribution and I have only some minor comments, suggestions and questions.

Line 85: What is meant with asynchronous (explain)?

Line 90: I think there does exist (yet probably not too detailed) kinematics of crawling (walking) in frogs (e.g. Nauwelaerts & Aerts, 1992; J. Zool.).

Line 134: This is a rather arbitrary definition of transition between swing and stance and vice versa, which is fine as such, but probably more prone to errors in its determination than the rigorous and most often used 'first contact' and 'last contact' of the hands and feet. Maybe give a short argumentation why this definition is used.

Line 173: Define 'body side' (as a segment) or at least refer explicitly to figure 1b.

Line 175: I find the use of 'V' to indicate (some) vectors rather confusing (at first I thought this referred to speed ('V'elocity). Why not using conventional vector script? (see also further).

Line 197: New notations are used for the same vectors.

Line 222: Increase length of the square root symbol (must encompass Z^2).

Line 223&226: I suggest to use another variable name. Presently flexion/extension angles are identical to joint angles (line: 197)

Line 251: This must be 'contribution of the pelvis' (in model 2 the hip also contributes because of femur rotation)

Line 251-258: Personally, I do not think this explanation is really needed.

Line 277: Judging what follows, M must be replaced by MT.

Line 282: use matrix notation and remove capitals in Cos and Sin

Line 284: remove '.' between R1 and MT. Moreover, I wonder whether the notation of equation 14 is correct (most likely calculations are OK). At first glance, R1 must not be transposed. Furthermore, notice that according this equation, MBA becomes a 3x12 matrix. As such, the explanation in lines 306-312 becomes very confusing (rotation of MBA). This means that MBA must not be transposed for the rotation, although this is done according to equation 18. On the other hand, this equation refers to MBAT, a matrix that is further not defined but probably means 'the transposition of MBA'. As said, this is all very confusing and in general I think it will be beneficial of the authors consult a mathematician to check and clean up the mathematical script (including the vector notations).

Line 385: sounds a bit odd. What is exactly meant here?

Line 410: 0.04m/s => 0.26m/s is a 6fold increase (not 3fold) and seems to accord to what can be seen in figure 6A

Line 411: If stride length is not changing, frequency must increase with a factor similar to that of speed (i.e. 6-fold; cf. above). This is confirmed by the frequency range in figure 6A (0.5hz =>3hz). Hence, the 3fold and 10fold increases mentioned in this part are confusing/puzzling.

Line 508: 6fold? (cf.above).

Reviewer: 2

Comments to the Author(s)

Review: The impact of pelvic lateral rotation on hindlimb kinematics and stride length in the red-legged running frog, *Kassina maculata*.

This study uses a clever non-invasive computational method to quantify the contribution of lateral pelvic movements to the stride length of walking frogs. The analysis is non-trivial yet done carefully and explained well and succinctly. Pelvic anatomy and function in frogs is a research topic that remains largely unexplored and this study will be a significant contribution. Overall, I really enjoyed the manuscript and would like to see it published. However, I would like to suggest some edits that I think would greatly improve the impact of this study.

First, I was hoping for a maximum calculated stride length increase with each degree of pelvic rotation, based purely on the geometry of the skeleton (Fig. 1B). Given how short and rigid the spine of the frog is I would not expect there to be much bending near the pelvis. Furthermore, the pelvis itself is narrow and so even large angular movements of the could conceivably have a small effect on stride length. So I was hoping for a hypothetical angle calculate, in the best case scenario, i.e. all joint extensions occur synchronously so that proximal rotation is fully transmitted to the distal limb.

The other thing I expected to see in the discussion was a constraint on stride length posed by the shorter forelimbs. For example, if the forelimb joints are near full extension during the stride an increase in hindlimb stride length would not be possible because the forelimbs would not be able to keep up.

Secondly, it was unclear for much of the manuscript what the two models were that the authors compared to each other. This was first described on line 238, even though there were several allusions to two approaches earlier (E.g. line 88, 162). I would have liked a brief description even in the abstract. At the end of the introduction (line 87) the first model is briefly described but the comparison remained incomplete.

Specific remarks

Line 22: IS - I would spell out this anatomical structure for the sake of those who are not frog anatomy fanatics (I don't know who these would be, but one never knows...!)

Line 24: range of speeds – units missing

Line 27: The reporting of results could use some more clarification. The last three sentences of the summary only made sense after reading the whole manuscript.

Line 68: "From salamander studies" could be omitted.

Line 70-71: I could find the illustration of this point in Figure 1.

Line 73-80: I had trouble seeing how pelvic rotation could not contribute to stride length increase.

Line 77-78: is this muscle activation pattern (citation 6) measured during walking or frog jumping?

Line 87: "The first model..." there is no follow up with the second model.

Line 162: "... using two methods" this made we wonder whether these were the two models being compared.

Line 176: "in terms of the skin marker landmarks". I expected some guidance on where to find the names for the landmarks that are given in the next page. It wasn't obvious what "P" in Pkne stood for...

Line 229-235: Excellent paragraph explaining the benefits of using polar coordinates!

Line 237-240: This was the first description of the 2 models. I would have liked to see it much earlier in the manuscript. The authors do a good job of explaining the two models succinctly so it should not be a problem to introduce them earlier.

Line 251-258: I thought the authors did a terrific job explaining global vs local coordinate systems before line 251. The further explanation with a different example (throwing a ball) could be omitted.

Line 329-330: I would have liked a little more explanation of why a sinusoidal motion in PV was expected. Is that due to the periodicity of the measurements? How can that be achieved if only one stride per sequence is digitized? I think this is a valuable contribution of this manuscript and strengthens the dataset despite the relatively small number of strides digitized. It is also not a frequently done analysis, even though it certainly should be. In my opinion, it therefore warrants a little more description.

Line 357: Is the upper limit of the speed observed in this study (0.23 m s⁻¹) close to the maximum for this species? At what speed does this frog species switch to hopping? This is perhaps a point for the discussion.

Line 368: "...creating sinusoidal curves" these are not the same sinusoidal waves as expected in the tests for digitizing error, correct?

Line 377: "...contributed only modestly to limb segment motion". This result was not obvious to me from the reference to figure 3D-I in line 375.

Line 404-406: I did not understand the difference between Vpel and lateral rotation, so I could not see how they had opposite relationship with speed (Fig. 5 vs 6).

DISCUSSION

Overall I had a hard time following the discussion to the final conclusion. There were often large jumps from some exceptional descriptions of complex 3-dimensional kinematics (e.g. line 467-478) to conclusions that did not follow directly.

Line 430-448: This is where I expected a discussion about the differences in pelvis length or structure among the different species. A similar anatomical comparison would be very useful in the following section on other sprawling tetrapods, as well.

Line 491: "... to 3.78 mm hypothetical increase in stride length" what is the origin of this hypothetical value?

Line 499-505: Great explanation and discussion of the lateral COM movements and the potential contribution to instability.

Line 517-520: In this section I expected discussion of the gait transition in *Kassina*, from walking to jumping.

Line 533-535: This sentence seemed contradictory within itself.

Author's Response to Decision Letter for (RSOS-190060.R0)

See Appendix A.

Decision letter (RSOS-190060.R1)

03-Apr-2019

Dear Dr Collings,

I am pleased to inform you that your manuscript entitled "The impact of pelvic lateral rotation on hindlimb kinematics and stride length in the red-legged running frog, *Kassina maculata*" is now accepted for publication in Royal Society Open Science.

on behalf of Professor Emily Standen (Associate Editor) and Kevin Padian (Subject Editor)
openscience@royalsociety.org

Appendix A

Associate Editor Comments to Author (Professor Emily Standen):

Dear Dr. Collings,

Thank you for the submission of your paper entitled 'The impact of pelvic lateral rotation on hind limb kinematics and stride length in the red-legged running frog, *Kassina maculata*. This is a very nicely written paper and an interesting approach.

The reviews are generally very positive about this paper but I feel as though they do have some comments that are worth considering. I look forward to seeing your revised manuscript.

Sincerely,

Emily Standen

Dear Professor Emily Standen, reviewers, and editorial staff,

We would like to thank you for your time reviewing and handling our manuscript.

Thank you to our subject and associate editor and reviewers for their helpful comments and suggestions. Additionally, many thanks goes to those at the editorial office for granting us a small extension enabling us the time to focus our full attention on our reviewer comments.

Our responses to every comment made can be found in red text below the original comment.

Best,

Dr Amber J Collings and Co-authors

Reviewer comments to Author:
Reviewer: 1

Comments to the Author(s)
Dear Editor,

I have considered the manuscript by Collings et al. on 'The impact of pelvic lateral rotation on hindlimb kinematics and stride length in the red-legged running frog, *Kassina maculata*'. Generally spoken, I like this contribution and I have only some minor comments, suggestions and questions.

Thank you!

Line 85: What is meant with asynchronous (explain)?

[R1.1] We originally used the word asynchronous here to note that the frogs when walking move their right and left hindlimbs asynchronously i.e. not in sync as they do in jumping. However upon reflection it is probably an unnecessary addition to the sentence and as such we have deleted it for clarity. Line 93.

Line 90: I think there does exist (yet probably not too detailed) kinematics of crawling (walking) in frogs (e.g. Nauwelaerts & Aerts, 1992; J. Zool.).

[R1.2] We agree, there are examples of walking studies including some kinematics, particularly the recent publication by Reynaga et al. (2018). Previous literature has also looked at the pelvis in isolation but we look at the impact of pelvic kinematic in relation to the hindlimb, and to our knowledge are the first to do so. We have reworded the sentence to clarify that this is the first use of 3D kinematics to analyse the pelvis in relation to the hindlimb. Line 100.

Line 134: This is a rather arbitrary definition of transition between swing and stance and vice versa, which is fine as such, but probably more prone to errors in its determination than the rigorous and most often used 'first contact' and 'last contact' of the hands and feet. Maybe give a short argumentation why this definition is used.

[R1.3] We do agree the definitions do seem quite arbitrary between swing and stance however it was extremely challenging to rely on the more traditional first and last contact. As we mention, the frogs gradually peeled their toes off of the ground and have a tendency to drag their long toes along the ground during swing phase which meant that in some instances there was no clear point of last contact at all even though the limb was protracting. Additionally, because of the elongated nature of their phase transitions there are periods of contact with the ground where weight bearing is minimal as thus for all intents and purpose the limb is effectively in swing. Similarly, the point of first contact with the first finger in the forelimb is very early on in the transition to stance phase and therefore in contact with the ground long before any weight bearing is going to occur. We appreciate that the arbitrary definitions we use are somewhat subjective but we believe our definitions are more useful than relying on first and last contact which just doesn't work for these animals that have such long digits! Lines 144-149 explain our reasoning for choosing arbitrary transition points. We have also added to the sentence starting line 149 to emphasise this is our best definition and is based on our chosen transition points. Line 149.

Line 173: Define 'body side' (as a segment) or at least refer explicitly to figure 1b.

[R1.4] We have now added a written definition of the joint segments and how they were calculated as vectors including for the body side. Line 192.

Line 175: I find the use of 'V' to indicate (some) vectors rather confusing (at first I thought this referred to speed ('V'elocity)). Why not using conventional vector script? (see also further).

[R1.5] Yes, this is a good point, thank you for raising this. We have corrected all uses of 'V' with '**v**' to denote vectors. Line 192 and throughout.

Line 197: New notations are used for the same vectors.

[R1.6] We can understand how our use of general symbols might be confusing; the equation is a generic equation regardless of the names used for any particular vector. However, to clarify, we have replaced the general symbols with vector names from above as an example of how hip 3D angle is calculated. We also explain that other joint angles are calculated similarly. Line 215 and 219.

Line 222: Increase length of the square root symbol (must encompass Z^2).

[R1.7] Yes, thank you for catching that. The square root symbol has been amended to include 'z²'. Line 247.

Line 223&226: I suggest to use another variable name. Presently flexion/extension angles are identical to joint angles (line: 197)

[R1.8] We have now changed the variable name for the flexion/extension angle from θ to σ to avoid confusion. Line 248 and throughout.

Line 251: This must be 'contribution of the pelvis' (in model 2 the hip also contributes because of femur rotation)

[R1.9] Yes you are correct, we have corrected 'hip' to 'pelvis'. Line 292.

Line 251-258: Personally, I do not think this explanation is really needed.

[R1.10] We have now removed this explanation. Line 278.

Line 277: Judging what follows, M must be replaced by MT.

[R1.11] Thank you for pointing this out. If M is in the 12x3 configuration then it needs to be transposed before being multiplied with R_1 . We have added an explanation to clarify. Line 324.

Line 282: use matrix notation and remove capitals in Cos and Sin

[R1.12] We have now added matrix notation and have removed the capital letters from cos and sin. Line 320 and 351.

Line 284: remove '.' between R1 and MT. Moreover, I wonder whether the notation of equation 14 is correct (most likely calculations are OK). At first glance, R1 must not be transposed. Furthermore, notice that according this equation, MBA becomes a 3x12 matrix. As such, the explanation in lines 306-312 becomes very confusing (rotation of MBA). This means that MBA must not be transposed for the rotation, although this is done according to equation 18. On the other hand, this equation refers to MBAT, a matrix that is further not defined but probably means 'the transposition of MBA'. As said, this is all very confusing and in general I think it will be beneficial of the authors consult a mathematician to check and clean up the mathematical script (including the vector notations).

[R1.13] You are correct R1 is not transposed and multiplication of R1 and M^T produces a 3x12 matrix. In our code workflow, M is a 12 x 3 matrix, M^T then is a 3 x 12 matrix which is multiplied with R1, to produce a 3 x 12 matrix. Our code requires all matrices used in downstream calculations to be in the 12 x 3 configuration. Therefore, the resulting matrix generated from the multiplication of R1 and M^T requires transposing itself, hence the notation $[R_1.M^T]^T$ (I have highlighted the extra transposition acting on the resulting matrix). M_{BA} then, being the transpose of the calculation within the matrix, is a 12 x 3 matrix as required for our code. See the equation below for further clarification:

$$RM = [R.M^T]^T$$

Where M is 12 x 3 and R is 3 x 3, producing RM as a 12 x 3 matrix.

Of course without the reader knowing how the matrix needed to be structured, we can see where the confusion has come from. To remain consistent with our code we have kept our notation the same however to help clarify we have reworded our text (Line 312-324 & 347-355) and added a line below the

equations in question defining dimensions (Line 324 & 355). We have also removed the '.' in the equations as suggested (Lines 322 & 353). Lines 312 – 355.

Line 385: sounds a bit odd. What is exactly meant here?

[R1.14] In this sentence we meant to highlight that although the pelvic lateral rotation angles reported for each trial were variable (i.e. some frogs showed a large pelvic angular excursion and others showed a smaller range of angles peak-to-trough) all traces were statistically significant when compared with signal noise – in our case digitising error.

We have now reworded the relevant sentences for clarification. Line 432.

Line 410: 0.04m/s => 0.26m/s is a 6fold increase (not 3fold) and seems to accord to what can be seen in figure 6A

[R1.15] Thank you for catching that. We have now corrected this typo. Line 462.

Line 411: If stride length is not changing, frequency must increase with a factor similar to that of speed (i.e. 6-fold; cf. above). This is confirmed by the frequency range in figure 6A (0.5hz =>3hz). Hence, the 3fold and 10fold increases mentioned in this part are confusing/puzzling.

[R1.16] Yes, you are absolutely correct, apologies! We have corrected these typos. Line 463.

Line 508: 6fold? (cf.above).

[R1.17] Again you are correct, thank you for spotting. This has now been corrected. Line 585.

Reviewer: 2

Comments to the Author(s)

Review: The impact of pelvic lateral rotation on hindlimb kinematics and stride length in the red-legged running frog, *Kassina maculata*.

This study uses a clever non-invasive computational method to quantify the contribution of lateral pelvic movements to the stride length of walking frogs. The analysis is non-trivial yet done carefully and explained well and succinctly. Pelvic anatomy and function in frogs is a research topic that remains largely unexplored and this study will be a significant contribution. Overall, I really enjoyed the manuscript and would like to see it published. However, I would like to suggest some edits that I think would greatly improve the impact of this study.

Thank you for your kind words.

First, I was hoping for a maximum calculated stride length increase with each degree of pelvic rotation, based purely on the geometry of the skeleton (Fig. 1B). Given how short and rigid the spine of the frog is I would not expect there to be much bending near the pelvis. Furthermore, the pelvis itself is narrow and so even large angular movements of the could conceivably have a small effect on stride length. So I was hoping for a hypothetical angle calculate, in the best case scenario, i.e. all joint extensions occur synchronously so that proximal rotation is fully transmitted to the distal limb.

[R2.1] As we understand, the reviewer is suggesting that the morphology of the pelvis might influence its hypothetical contribution to stride length. Specifically, the width of the pelvis might prevent the femur from reaching a highly protracted angle. This is an interesting point which, as the reviewer admits, is dependent on an assumption about spinal flexibility. Unfortunately, we cannot reliably integrate this into our current calculations because we did not perform x-ray fluoroscope experiments and therefore were unable to observe a) the precise morphology of the pelvis for each frog b) the flexion of the spine. We have added a comment in our discussion that this issue should be addressed using XROMM experiments in the future. Line 608.

The other thing I expected to see in the discussion was a constraint on stride length posed by the shorter forelimbs. For example, if the forelimb joints are near full extension during the stride an increase in hindlimb stride length would not be possible because the forelimbs would not be able to keep up.

[R2.2] Yes, this is a very good point. Indeed, the legs of *Kassina* are actually shorter to minimise this effect (Reynaga et al., 2018). We have addressed this in our text. Line 579.

Secondly, it was unclear for much of the manuscript what the two models were that the authors compared to each other. This was first described on line 238, even though there were several allusions to two approaches earlier (E.g. line 88, 162). I would have liked a brief description even in the abstract. At the end of the introduction (line 87) the first model is briefly described but the comparison remained incomplete.

[R2.3] Yes, we agree, without clear description in the introduction it was unclear what the two models were. Model 1 was called the fixed body axis model (FBA) and model 2 was called the fixed pelvic axis model (FPA). We have now added sentences and headings in the various relevant sections to clarify the definition of the two models. Line 24, Line 32, Line 95, Line 269, Line 265, Line 274, Line 290, Line 294, Line 332, Line 361, Line 362.

Specific remarks

Line 22: IS - I would spell out this anatomical structure for the sake of those who are not frog anatomy fanatics (I don't know who these would be, but one never knows...!)

[R2.4] haha! Yes very good point! We have now modified that sentence for clarity. Line 25.

Line 24: range of speeds – units missing

[R2.5] Ah yes, thank you for catching that. The correct units have now been added. Line 28.

Line 27: The reporting of results could use some more clarification. The last three sentences of the summary only made sense after reading the whole manuscript.

[R2.6] changed sentence structure to help clarify the results. Line 28 onwards.

Line 68: "From salamander studies" could be omitted.

[R2.7] deleted. Line 75.

Line 70-71: I could find the illustration of this point in Figure 1.

[R2.8] Apologies for this typo, we did not mean to refer to Figure 1 there. We have also re-worded that sentence for further clarification. Line 78.

Line 73-80: I had trouble seeing how pelvic rotation could not contribute to stride length increase.

[R2.9] lateral rotation of the pelvis will only affect stride length if 1) rotation is large enough, 2) limb and pelvic motion are temporally synchronised – depending on the synchronisation pelvic lateral rotation can have a positive or negative effect on stride length. We discuss this in more detail in the discussion between Lines 535-545.

Line 77-78: is this muscle activation pattern (citation 6) measured during walking or frog jumping?

[R2.10] The work done by Emerson and De Jongh (citation 6) looks at muscle activity during walking, jumping, and swimming and has the EMG data for the coccygeoiliacus, coccygeosacralis, iliolumbaris, and longissimus dorsi. We have clarified we are talking about muscle action in walking. Line 85.

Line 87: "The first model..." there is no follow up with the second model.

[R2.11] You are right, thank you for bringing that to our attention. As with comment [R2.3] We have now added a description of the second model to this section for clarification. Line 95.

Line 162: "... using two methods" this made we wonder whether these were the two models being compared.

[R2.12] We appreciate where the confusion has arisen. To clarify, the hindlimb kinematics were analysed using both 3D vector angles and a polar co-ordinate angle. However, the models being compared were the fixed body axis kinematic model (FBA) and the fixed pelvic axis kinematic model (FPA). FBA was fixed to the body axis and therefore measured pelvic excursion *and* hindlimb motion, whereas FPA was fixed to the pelvic axis and therefore measured hindlimb motion *only* without the contribution of the pelvis. The changes we made in response to comment [R2.3] hopefully clarify this point, see Line 24, Line 32, Line 95, Line 269, Line 265, Line 274, Line 290, Line 294, Line 332, Line 361, Line 362.

Line 176: "in terms of the skin marker landmarks". I expected some guidance on where to find the names for the landmarks that are given in the next page. It wasn't obvious what "P" in Pkne stood for...

[R2.13] We used 'P' to stand for points, as in joint marker points. We felt that 'M' for markers was too confusing since we introduce matrices later on that are referred to by the letter 'M'. We have re-written this section and added some new text to define the marker points, and directed readers to Figure 1 A-B where they can find a labelled diagram of the frog showing the marker points and the vectors they create. Line 182 onwards.

Line 229-235: Excellent paragraph explaining the benefits of using polar coordinates!

[R2.14] Thank you very much!

Line 237-240: This was the first description of the 2 models. I would have liked to see it much earlier in the manuscript. The authors do a good job of explaining the two models succinctly so it should not be a problem to introduce them earlier.

[R2.15] Thank you, we agree. Please see all revisions labelled [R2.3]. Line 24, Line 32, Line 95, Line 269, Line 265, Line 274, Line 290, Line 294, Line 332, Line 361, Line 362.

Line 251-258: I thought the authors did a terrific job explaining global vs local coordinate systems before line 251. The further explanation with a different example (throwing a ball) could be omitted.

[R2.16] Thank you very much. We have now deleted the analogy. Line 278.

Line 329-330: I would have liked a little more explanation of why a sinusoidal motion in PV was expected. Is that due to the periodicity of the measurements? How can that be achieved if only one stride per sequence is digitized? I think this is a valuable contribution of this manuscript and strengthens the dataset despite the relatively small number of strides digitized. It is also not a frequently done analysis, even though it certainly should be. In my opinion, it therefore warrants a little more description.

[R2.17] We know from the work of Ashley-Ross (Fig 4 citation 29) that the pelvis in salamanders oscillates sinusoidally during walking/running and given we were trying to measure a similar motion in the frogs (i.e. pelvic lateral rotation) we expected a similar sine wave pattern. As for whether you can get a sine wave with one stride, one must make the assumption that the pattern repeats over multiple stride cycles. Since, again, this is true of salamander pelvic oscillation we assume that it is also the case in the frogs. We feel comfortable with this assumption for our current data because we put great effort into ensuring we only used strides that represented steady speed locomotion where the animal was travelling forwards in a straight line i.e. not pausing or turning. We have now added text to explain the rationale for our expectation of oscillating pelvic angle in both the methods and the discussion. Line 373 and Line 484, respectively.

Line 357: Is the upper limit of the speed observed in this study (0.23 m s^{-1}) close to the maximum for this species? At what speed does this frog species switch to hopping? This is perhaps a point for the discussion.

[R2.18] As far as we are aware, yes, this is approaching the upper limit of recorded running speeds for them. There is not that much literature for this particular species however Ahn et al. (2004) has reported speeds between 0.10 m s^{-1} and 0.30 m s^{-1} . The great thing about this species – and indeed one of the reasons we chose to work with it for this study – is that they are not known to switch to hopping at all. Anecdotally in our experience with them they perform a single escape jump, a short burst of hopping strides, or a series of walking strides, they never switch. Ahn et al. (2004) also make note of this characteristic explaining that “*Kassina* uses this walking/running-type gait as its primary mode of locomotion at all speeds on land”. We discuss their ability to maintain a walking/running at all speeds without transitioning to another gait between Lines 585 and 599. We have added a new sentence in this section to clarify that this species don’t switch gaits. Line 593.

Line 368: “...creating sinusoidal curves” these are not the same sinusoidal waves as expected in the tests for digitizing error, correct?

[R2.19] No, that is correct, in line 415 (originally 368) we are referring to the hindlimb joint angle traces. We tested the pelvic angle traces for digitising error since the pelvic angles were undergoing a much smaller range of motion. In accuracy the pelvic angle traces aren’t strictly sinusoidal, they are

more triangular (like a zig zag wave rather than a smooth 'S' shaped curve) so on reflection we have decided to refer to the pelvic angle traces as triangular waves and the hindlimb joint angle traces as sinusoidal waves. We have amended our text accordingly. Line 431.

Line 377: "...contributed only modestly to limb segment motion". This result was not obvious to me from the reference to figure 3D-I in line 375.

[R2.20] We say this because the traces in Figures 3D-F show a fluctuation in horizontal angle throughout the stride cycle indicating the limb segments are moving backwards and forwards throughout the limb cycle. On the other hand figure 3G-I show very little change in vertical angle throughout the stride cycle suggesting the limb segments don't move up and down very much during a stride cycle hence our statement that vertical motion contributed only modestly to overall limb segment motion. We have re-worded the sentence slightly to help clarify this point. Line 424.

Line 404-406: I did not understand the difference between Vpel and lateral rotation, so I could not see how they had opposite relationship with speed (Fig. 5 vs 6).

[R2.21] Vpel is the pelvic vector itself, the angle of lateral rotation (PV) is the angle between the midline and Vpel. You are correct, they have the same relationship with speed as each other. Lateral rotation is correlated with an increase in the cranio-caudal excursion of the hindlimb joints, but has a negative relationship with speed. We have re-worded this section slightly to help clarify. Line 453 onwards.

DISCUSSION

Overall I had a hard time following the discussion to the final conclusion. There were often large jumps from some exceptional descriptions of complex 3-dimensional kinematics (e.g. line 467-478) to conclusions that did not follow directly.

[R2.22] We have reworked the second half of the section entitled 'Pelvic lateral rotation only modestly increases stride length and decreases with speed' to help with the flow of discussion points. On reflection there was a lot of unnecessary repetition of the previous points, so we have deleted text to better link with the following paragraph. Line 566.

Line 430-448: This is where I expected a discussion about the differences in pelvis length or structure among the different species. A similar anatomical comparison would be very useful in the following section on other sprawling tetrapods, as well.

[R2.23] We have added to this paragraph now to include looking at the impact of pelvic geometry on hindlimb kinematics as a suggestion for future work. Line 499.

Also see our response to comment [R2.1]. Line 608.

Line 491: "... to 3.78 mm hypothetical increase in stride length" what is the origin of this hypothetical value?

[R2.24] The hypothetical gain in stride length was calculated by comparing the cranio-caudal excursion of the joints in Model 1 and Model 2. Figure 4B-C provides a visual representation of how hypothetical gain was calculated. The value of 3.78 is actually a typo and should read 3.72 which is taken from Table S2. We have corrected the typo, have moved Tables 1 and 2 out of supplementary information and into the main text, and have then made reference to Table 2 in the relevant sentence. Line 562.

Line 499-505: Great explanation and discussion of the lateral COM movements and the potential contribution to instability.

[R2.25] Thank you very much for your kind words.

Line 517-520: In this section I expected discussion of the gait transition in *Kassina*, from walking to jumping.

[R2.26] We have added a sentence to clarify that *Kassina maculata* does not switch gaits. Line 593.

Line 533-535: This sentence seemed contradictory within itself.

[R2.27] You are right, what was meant is that if the animals were not able to laterally rotate their pelvis they would compensate if necessary which might impact stride length. We are making the point that we are not saying our model shows exactly what would happen if you anatomically fixed the pelvis but rather that our model is being used to demonstrate that rotation of the pelvis has the potential to impact stride length (although it turns out just not all that much!). We have slightly re-worded the sentence to help clarify. Line 614.